# Apolipoprotein B/Apolipoprotein A-I Ratio Is a Better Predictor of Cancer Mortality Compared with C-Reactive Protein: Results from Two Multi-Ethnic US Populations

**DOI:** 10.3390/jcm9010170

**Published:** 2020-01-08

**Authors:** Mohsen Mazidi, Niki Katsiki, Dimitri P. Mikhailidis, Dina Radenkovic, Daniel Pella, Maciej Banach

**Affiliations:** 1Department of Twin Research and Genetic Epidemiology, King’s College London, St Thomas’ Hospital, Strand, London SE1 7EH, UK; 2First Department of Internal Medicine, Center for Diabetes, Metabolism and Endocrinology, AHEPA University Hospital, 546 36 Thessaloniki, Greece; nikikatsiki@hotmail.com; 3Department of Clinical Biochemistry, Royal Free Campus, University College London Medical School, University College London (UCL), London NW3 2QG, UK; MIKHAILIDIS@aol.com; 4Guy’s & St Thomas’ NHS Foundation Trust, London SE1 7EH, UK; dina.radenkovic@gmail.com; 52nd Cardiology Clinic East Slovak Institute for CV Disease and Faculty of Medicine PJ Safarik University, 04011 Kosice, Slovakia; daniel.pella@upjs.sk; 6Department of Hypertension, Chair of Nephrology and Hypertension, Medical University of Lodz, 93-338 Lodz, Poland; 7Polish Mother’s Memorial Hospital Research Institute (PMMHRI), 93-338 Lodz, Poland; 8Cardiovascular Research Centre, University of Zielona Gora, 65-046 Zielona Gora, Poland

**Keywords:** apolipoprotein B, apolipoprotein A-I, C-reactive protein, cancer mortality, NHANES

## Abstract

Background: There is a lack of evidence regarding the link between apolipoproteins and cancer mortality. By using two nationally representative samples of US adults, we prospectively evaluated the associations between apolipoprotein B (apoB) levels and apoB/apoA-I ratio with cancer mortality. We also examined the role of C-reactive protein (CRP) in these associations. Materials and Methods: Adults aged ≥20 years, enrolled in the 3rd National Health and Nutrition Examination Survey (NHANES III, 1988–1994) and continuous NHANES (2005–2010), and followed up to 31 December 2011, were included in the analysis. Multiple Cox regressions were applied to evaluate the associations between the variables of interest and cancer mortality. Results: Overall, 7695 participants were included (mean age: 49.2 years; 50.4% men, median follow-up: 19.1 years). In the fully adjusted model, participants in the highest quartile (Q4) of apoB/apoA-I had a significantly greater risk for cancer mortality (hazard ratio (HR): 1.40; 95% confidence interval (CI): 1.25–1.93) compared with those in the first quartile (Q1). In the same model, a positive and significant association between apoB levels and cancer mortality was observed for individuals in Q3 (HR: 1.12; 95% CI: 1.09–1.16) and Q4 (HR: 1.17; 95% CI: 1.09–1.25) compared with those in Q1. When CRP levels were added in the analysis, the apoB/apoA-I ratio, but not apoB levels, remained significantly related to cancer mortality (Q4 = HR: 1.17; 95% CI: 1.09–1.25). In contrast, CRP levels were not able to predict cancer death after correction for apoB/apoA-I ratio. Conclusions: In a large representative sample of the US adult population, the apoB/apoA-I ratio and apoB levels significantly predicted cancer mortality, independently of several cardiometabolic risk factors. The predictive value of apoB/apoA-I, but not apoB levels, remained significant after taking into account CRP, whereas CRP was not associated with cancer mortality after adjustment for apoB/apoA-I ratio. If further evidence supports our findings, apoA-I and apoB measurements could be considered in general healthcare policies.

## 1. Introduction

According to the World Health Organization (WHO), 7.6 million people worldwide died from malignant neoplasms (13% of all deaths) in 2008, and this number is predicted to rise to 12 million by 2030 [1]. Given this burden of morbidity and mortality, reducing the risk for developing cancer could be considerably beneficial.

Apolipoprotein A-I (apoA-I), the major protein component of high-density lipoprotein (HDL), is synthesized predominantly in the liver and small intestine [2,3]. ApoA-I particles transport excess cholesterol from peripheral tissues to the liver, while they also exert anti-inflammatory, anti-apoptotic and antioxidant actions [2,4,5]. In contrast, apoB-containing lipoproteins are atherogenic [6]. Recent prospective studies indicated that the apoB/apoA-I ratio might be a useful predictor of the risk for cardiovascular disease (CVD) mortality [7]. It was also shown that this ratio may even better predict CVD risk than lipid biomarkers [8]. However, data are still conflicting [9]. In this regard, there is an ongoing debate as to whether apo measurements should be included in clinical guidelines [9,10,11].

It has been also suggested that apoA-I could play a protective role against cancer [12]. However, studies on the link between cancer and apoA-I are controversial and still relatively rare. Reduced plasma levels of apoA-I have been identified in patients with early-stage ovarian cancer compared with normal individuals [12]. Similarly, serum apoA-I levels were found to be 2-fold lower in patients undergoing surgery for pancreatic cancer compared with healthy controls [13]. Lower serum apoA-I levels were also associated with a higher risk of breast cancer [14] and its recurrence [15], although other studies showed no such correlation [16] or even an inverse association [17]. Another study reported a positive link between serum apoA-I concentrations and breast cancer risk [18]. The association of apoB with cancer is far less clear, since apoB studies have mostly focused on cardiometabolic disorders [19,20]. In this context, elevated serum apoB levels have been related to diabetes and metabolic syndrome [19,20], both of which may affect cancer development [21]. Furthermore, there are reports linking increased apoB levels with a higher risk for lung and colorectal cancer, as well as low apoB concentrations with a greater risk for breast cancer [22].

C-reactive protein (CRP), an acute-phase plasma protein, is one of the most frequently used inflammatory markers. Since inflammation is recognized as a critical component of tumor progression [23], CRP could be a predictor of cancer mortality [24,25]. However, it remains unclear whether apo levels affect the associations between CRP and cancer.

Based on the above, in the present study, data from two large, population-based, prospective US studies were analyzed to evaluate any links between apoB levels and the apoB/apoA-I ratio with cancer mortality. We also investigated whether CRP could affect the predictive potential of the apoB/apoA-I ratio for cancer mortality and vice versa.

## 2. Materials and Methods

### 2.1. Study Population

The 3rd National Health and Nutrition Examination Survey (NHANES III) (1988–1994) is a multistage stratified survey designed to provide a detailed examination of the health and nutritional status of a nationally representative sample of non-institutionalized individuals in the US [26]. The study protocol was approved by the institutional review board at the Centers for Disease Control and Prevention (Atlanta, GA, USA), and written informed consent was obtained from all participants. Baseline data in the NHANES III were collected during household interviews; blood and urine samples were provided during a medical examination visit. Demographic information—including age, sex, ethnic origin, household income, education, and smoking status—was obtained during the household interview. Information on body mass index (BMI), physical activity, blood pressure, and diet were obtained during the medical examination visit. Among 18,714 adults aged ≥20 years in the NHANES III, we excluded adults who reported a previous history of CVD and cancer (*n* = 2869) and those currently pregnant (*n* = 287) as well as 7863 participants with missing data for the apoB/apoA-I ratio. Overall, 7695 adults were included in the present analysis.

To externally validate our results for the link between apoB and CRP levels with cancer mortality, the NHANES continuous dataset was used (2005–2011). Further detailed information on the methodology used in the NHANES is published elsewhere [27,28]. Continuous NHANES is an ongoing yearly survey designed to assess the health and nutritional status of the general, non-institutionalized US population [27,28]. The National Center for Health Statistics (NCHS) Research Ethics Review Board approved the underlying protocol, and written informed consent was obtained from all participants. Details on the NHANES Laboratory/Medical Technologists and Anthropometry Procedures are described elsewhere [27,28].

### 2.2. Apolipoproteins Analysis

Samples were defrosted at room temperature and mixed thoroughly for 30 min on a blood-rotating device before analysis. ApoB and apoA-I were measured by radial immunodiffusion in the first 8.2% (1055 specimens) of the specimens during the first 5 months of the study and by rate immunonephelometry for the remaining specimens during the last 31 months. At the beginning of the survey, there were no standardized reference materials on which to base these measurements. Over the past few years, the WHO–International Federation of Clinical Chemistry (WHO-IFCC) First International Reference Materials for apoB and apoA-I became available. The Northwest Lipid Research Laboratories (Seattle, WA, USA) served as the coordinating laboratory for the development of these materials. The results were used to transform the immunonephelometry values to equivalent WHO-IFCC International Reference Materials-based values, which are presented here. More detailed methodology and laboratory procedures of the NHANES III are published elsewhere [29].

### 2.3. Mortality

A full description of mortality linkage methods is available from the NCHS. Briefly, the NCHS staff linked participants in the NHANES III to the underlying cause of death in the National Death Index (NDI) with a series of identifiers (e.g., social security number and date of birth) using probabilistic matching criteria. Individuals were followed up to 31 December 2011 (17- to 23-year follow-up, median follow-up: 19.1 years). If a match was not made with the NDI, that person was assumed to be alive as of that date. The underlying cause of death was obtained using codes from the International Classification of Diseases ninth (ICD-9; 1988–1998) or tenth (ICD-10; 1999–2006) version. For the NHANES continuous (2005–2011), mortality follow-up data are available from the date of survey participation until 31 December 2011. We examined mortality due to cancer codes (C00-C97). Cause of death was determined using the ICD-10.

### 2.4. Covariates

Blood pressure (in mmHg) was measured using a mercury sphygmomanometer while participants were in a seated position. Three measurements were recorded and then averaged for each individual to minimize measurement error. Hypertension (HTN) was defined as systolic blood pressure ≥140 mmHg or diastolic blood pressure ≥90 mmHg and/or use of antihypertensive medications. The Healthy Eating Index (HEI) includes 10 components: fruits, vegetables, grains, dairy, meats, fats, saturated fat, cholesterol, sodium, and dietary variety. The total score ranges from 0 to 100, with a higher score indicating a healthier diet. Smoking status was categorized as current, former, or never. Race/ethnicity (non-Hispanic white, non-Hispanic black, Mexican American, or other), educational attainment (<12, 12–15, ≥16 years of education), alcohol intake (never, 1–2, ≥3 drinks/week) were also recorded. Diabetes was defined as fasting plasma glucose of ≥126 mg/dL, current use of antidiabetic drugs or self-reported diagnosis of diabetes. Poverty-to-income ratio (PIR), a ratio of total family income to the official poverty threshold, was used to assess an individual’s socioeconomic status. Body mass index (BMI) was calculated as weight in kg divided by the square of height in m. CRP concentrations were measured by latex-enhanced nephelometry on a Behring Nephelometer (Dade Behring Diagnostics Inc., Somerville, NJ, USA). Triglycerides and cholesterol were measured enzymatically with commercially available reagents (Cholesterol/HP, cat. no. 816302, and Triglycerides/GPO, cat. no. 816370, both from Boehringer Mannheim, Indianapolis, IN, USA). Dyslipidemia was defined as any of the following: total cholesterol > 240 mg/dL (>6.21 mmol/L); low-density lipoprotein cholesterol (LDL-C) > 160 mg/dL (>4.14 mmol/L); high-density lipoprotein cholesterol (HDL-C) < 40 mg/dL (<1.03 mmol/L); or by the use of cholesterol-lowering medications.

### 2.5. Statistical Analysis

The analysis of the NHANES III data was conducted according to the guidelines included in the Analytic and Reporting Guidelines for NHANES III (1988–1994) [30]. Continuous and categorical demographic variables were compared across apoA quartiles using analysis of variance (ANOVA) and chi-squared tests, respectively.

We constructed Kaplan–Meier survival curves according to quartiles of apoB and the apoB/apoA-I ratio compared differences for the composite endpoints across groups using the log-rank test. All tests were two-sided and *p* < 0.05 was considered significant. Multivariable Cox proportional hazards were applied to determine the hazard ratio (HR) and 95% confidence interval (CI) of cancer mortality for apoB, apoB/apoA-I ratio, and CRP. Two different models were used, namely Model 1 (adjusted for age, sex, race, PIR, and education level) and Model 2 (Model 1 additionally adjusted for alcohol intake, HEI, BMI, smoking, dyslipidemia, HTN, and diabetes). To address whether the apoB/apoA-I ratio had incremental predictive utility over the CRP, we performed multivariable Cox proportional hazards regression to investigate the relations of the apoB/apoA-I ratio to cancer death adjusting for traditional risk factors and CRP. All statistical analyses were conducted using the R version 3.4.0 and SPSS^®^ complex sample module version 22.0 (IBM Corp., Armonk, NY, USA).

## 3. Results

### 3.1. Descriptive Characteristics

We included 7695 participants from the NHANES III with a mean ± standard error of mean (SEM) age of 49.2 ± 0.3 years; 50.4% were males. Table 1 shows the baseline participant characteristics. After a median follow-up of 19.1 years, 692 cases of cancer mortality were recorded in the total sample of the NHANES III. For the continuous NHANES, we included 7895 individuals (mean age: 49.7 ± 0.2 years; 51.8% females). After a median follow-up of 4.1 years, 402 cases of cancer death were recorded. Kaplan–Meier survival plots demonstrate that subjects in the lowest quartiles of apoB had significantly decreased cumulative cancer mortality (and better overall survival) than those in the highest quartiles (log-rank *p* < 0.0001), and similar patterns were observed for apoB/apoA-I ratio (log-rank *p* < 0.0001).

### 3.2. Relations of Apos and CRP with Cancer Mortality

Across the quartiles of apoB/apoA-I ratio in Model 1 (adjusted for age, sex, race, PIR, and education level), participants in the highest quartile (Q4) had a 62% greater risk of cancer mortality compared with the first quartile (Q1 = HR: 1.62; 95% CI: 1.38–1.78). After further adjustments (Model 2: age, sex, race, PIR, education level, alcohol intake, HEI, smoking, BMI, dyslipidemia, HTN, and diabetes), this link was slightly attenuated but it was still significant and positive (Q4 = HR: 1.40; 95% CI: 1.25–1.93) (Table 2).

Participants in the second, third, and fourth quartiles of apoB had 18%, 21%, and 26% higher risks of cancer death compared with the first quartile, respectively. After further adjustments in Model 2, positive and significant relations were observed between individuals in the third and fourth quartiles compared with Q1 (Q3 = HR: 1.12; 95% CI: 1.09–1.16; Q4 = 1.17; 95% CI: 1.09–1.25). Results from the continuous NHANES were similar; in Model 2, participants in the third (HR: 1.15; 95% CI: 1.08–1.23) and fourth (HR: 1.22; 95% CI: 1.10, 1.38) quartiles had greater risks of cancer mortality compared with the first quartile (by 15% and 22%, respectively). Furthermore, across the quartiles of CRP, individuals in the highest quartile had a significantly higher cancer death risk compared with the first quartile (Q4 = HR: 1.30; 95% CI: 1.16, 1.50). This relation was evident even after further correction in Model 2 (Q4 = HR: 1.22; 95% CI: 1.18, 1.28).

### 3.3. Predictive Comparisons

The apoB/apoA-I ratio remained significantly related to cancer mortality (HR: 1.17; 95% CI 1.09–1.25), even after adjusting for traditional risk factors (age, sex, race, alcohol and dietary intake, smoking, BMI, dyslipidemia, hypertension, and diabetes) and CRP. In contrast, CRP was not a significant predictor of cancer death (HR: 0.96; 95% CI 0.54–1.59), after adjustment for traditional risk factors (age, sex, race, alcohol intake, HEI, smoking, BMI, dyslipidemia, HTN, and diabetes) and the apoB/apoA-I ratio. Similarly, after adjustment for the apoB/apoA-I in the Cox model, apoB could no longer predict cancer mortality (HR: 1.04; 95% CI 0.96–1.13).

## 4. Discussion

Based on the obtained results, we showed that apoB/apoA-I ratio was significantly associated with cancer mortality, independently of other cardiometabolic risk factors. We also noticed that apoB levels might be effective in the prediction of cancer death, which was comparable with the apoB/apoA-I ratio. Finally, apoB/apoA-I, but not apoB, was a better predictor of cancer death than inflammatory markers (CRP) and other investigated traditional cardiometabolic risk factors. These findings suggest that the measurement of apos may have important clinical relevance in identifying subjects at risk for fatal cancer disease.

We found that individuals with higher apoB/apoA-I ratio and apoB levels might have a greater cancer mortality risk. These findings are in agreement with other studies, reporting either a protective role of apoA [12,13,14,15,16,17,18] or a detrimental impact of apoB [19,20,21,22] on cancer death. However, we need to highlight that studies on the link between apoA-I and apoB with cancer mortality have controversial results [12,13,14,15,16,17,18]. In this context, Kim et al. [12] measured apoA for evaluating the prognosis of ovarian cancer (with 61 healthy individuals, 84 pts with benign ovarian disease and 118 pts with ovarian cancer). ApoA was a good predictor of ovarian cancer, since patients with early-stage ovarian cancer had reduced plasma apoA levels compared with normal individuals [12]. In another study, the authors [13] aimed to identify new biomarkers in pancreatic cancer patients; they reported that apoA concentrations were decreased at least 2-fold in pancreatic cancer patients (*n* = 96) compared with the control group (*n* = 96 healthy volunteers) [13]. Sierra-Johnson et al. investigated Taiwanese women (150 breast cancer patients before treatment and 71 healthy controls), and showed that breast cancer patients had significantly lower apoA levels and apoA-I/apoB ratios than controls [11]. After adjusting for HDL-C and apoB/apoA-I ratio, breast cancer patients still had significantly lower apoA levels, thus highlighting a potential role of this apolipoprotein in breast cancer [11]. However, not all studies showed concordant results. In a cohort study, among 38,823 Norwegian women aged 17–54 years (follow-up: 17.2 years, 708 cases of invasive breast cancer identified), no link between apoA and postmenopausal breast cancer risk was found [16]. Furthermore, in the study by Martin et al. (involving 279 invasive breast cancer patients and 558 matched controls, followed for an average of 10 years), serum apoA-I concentrations were positively linked to breast cancer risk [18].

The associations of apoB levels with cancer have been previously evaluated [18,22]. In this context, the Malmö Diet and Cancer Study (*n* = 17,035 women and 11,063 men) showed that apoB was positively associated with overall cancer risk (HR: 1.06; 95% CI: 1.01–1.10) only in men, as well as with colorectal cancer risk (HR: 1.08; 95% CI: 1.01–1.16) in both genders [19]. In women, apoB was inversely related to breast cancer risk (HR: 0.92; 95% CI: 0.86–0.99), whereas in both genders, apoA-I was negatively linked to lung cancer risk (HR: 0.88; 95% CI: 0.80–0.97). Similarly, serum apoB levels were negatively associated with breast cancer risk in another nested case–control study [18]. Of note, apoB has been adversely linked to metabolic syndrome components and cardiovascular risk factors [19,20].

We showed that individuals with higher CRP levels at baseline had a greater risk of cancer mortality, in line with several other studies [23,24,25,31,32]. Allin et al. followed up with 10,408 individuals from the Danish general population for up to 16 years and found a 2- (HR: 2.2; 95% CI: 1.0–4.6) and 1.3-fold (HR: 1.3; 95% CI: 1.0–1.6) increased risk of lung cancer and cancer of any type in those in the highest vs. the lowest quartile of CRP concentrations [31]. A similar finding regarding lung cancer was reported in another prospective nested case–control study (*n* = 592) [32]. Furthermore, Otani et al. found that individuals in the highest quartile of CRP levels had a 1.6-fold (OR: 1.6; 95% CI: 1.1–2.5; *p*-trend = 0.053) greater risk of colorectal cancer compared with those in the lowest quartile [33], as supported also by a meta-analysis (1.12-fold increased risk per 1 unit change in natural log-transformed CRP levels) [34]. In contrast, Zhang et al. (*n* = 169) showed no association between elevated CRP levels and risk of colorectal cancer [35]. However, we need to mention that it is not clear whether CRP is a marker of cancer mortality or whether it may play a causative role [25].

In the present study, the apoB/apoA-I ratio was superior compared with CRP for predicting cancer mortality. To the best of our knowledge, this is the first study comparing the predictive value of the apoB/apoA-I ratio and CRP in terms of cancer death. The Mendelian randomization design may help in the random assortment of alleles from parents to offspring to provide a relatively unbiased assessment of whether exposures (in this case, CRP levels) are causally related to outcome. In this context, two available Danish studies (with *n* = 10,215, and *n* = 36,403, respectively) showed no association between CRP genotypes and risk of lung, colorectal, breast, prostate, bladder, or urinary tract cancer [25,36]. Similarly, a nested case–control study with 378 cases of lung cancer by Chaturvedi et al. showed no relationship between five CRP single-nucleotide polymorphisms (SNPs) and risk of lung cancer [32]. Another study by Heikkila et al. examined the association between CRP polymorphisms and risk of colorectal, lung, prostate, and breast cancer, as well as total cancers [37]. The authors genotyped almost 19,000 individuals from three population-based prospective cohort studies for 13 CRP-related SNPs that were identified from the genome-wide association study, and found a few positive associations of genetic polymorphisms with cancer risk. However, when they performed an instrumental variable analysis, it did not provide evidence for a causal link of CRP concentrations with cancer risk [32,36,37].

There are several plausible explanations for the association between apoA-I and cancer mortality. Preclinical studies indicated that apoA-I can exert anticancer properties due to the conversion of tumor-promoting M2 into anti-tumor M1 macrophages, the improvement of tumor infiltration by cytotoxic CD8 T cells, and reduced angiogenesis [38]. Furthermore, suppressed gene expression and/or apoA-I levels have been observed in breast cancer patients [14,38,39,40]. This may represent an independent prognostic factor for shorter disease-free survival and overall survival (OS) in patients with invasive ductal carcinoma [38,41]. Consistently, higher levels of apoA-I were inversely linked with the colon cancer risk [38,42]. Additionally, using a proteomic profiling approach, it was shown that decreased concentrations of three apoA-I specific peptides were related to the incidence of the liver cancer, thus potentially representing novel biomarkers for the early detection of hepatocellular carcinoma [38,43] Furthermore, apoA-I levels were inversely associated with the development of tumor metastases in hepatocellular carcinoma patients [44]. Among those with gynecological tumors, higher apoA-I mRNA expression was linked to better OS in ovarian carcinoma cases [45]. Owing to the pivotal role of apoA-I in reverse cholesterol transport, the above-mentioned findings may imply that increased apoA-I concentrations in the presence of cancer can enhance cholesterol efflux and reverse cholesterol transport and therefore exert an inhibitory effect on tumor cell proliferation and growth that have a high demand for cholesterol [46]. Some other explanations may be also found in the recent analyses and hypothesis-driven papers on the potential role of HDL cholesterol and cancer risk [47,48,49].

Low apoA/apoB ratio has been associated with increased lung and colorectal cancer risk, but reduced breast cancer risk in females [1]. Indeed, such serum markers may have different associations with different cancer types. For example, apoA1 seems protective for several malignancies, but detrimental for colorectal cancer [5]. Furthermore, studies have suggested relations of these apos with cancer prognosis. In this context, high apoB was related to increased intra-ocular metastasis in a Chinese population [2], whereas dysfunctional apoB mutations were associated with 46% increased risk of poor hepatocellular carcinoma outcomes [3]. Individual serum markers may also be useful as predictors of cancer development; for example, apoA2 may detect pancreatic cancer at an early stage [4].

The present study has some limitations. The relatively large sample size, long period of follow-up, and rigorous approach to data collection in the NHANES make this cohort a useful tool to explore relationships between investigated biomarkers and risk of mortality. However, due to its observational design, the possibility of influence by unmeasured or residual confounding cannot be excluded. Although our analysis was corrected for several covariates (due to data unavailability, some of the variables were not adjusted in the applied models, such as covariates for cancer screening, e.g., mammogram, colonoscopy, or a composite variable for cancer screening), we should mention that it would be better for future studies to consider more variables or use a different study design to minimize the impact of other factors. Other potential limitations could be the single measurement of apoB/apoA-I ratio, as well as the not completely clear stability of this biological marker over time. For future studies, it would be important to look at each cancer type separately, since different types of malignancies may have a varied association with apoB or apoA levels, as we discussed above.

The strengths of the present study include the use of long-term, large population-based data from two independent databases and the random selection of both the samples, thus allowing the generalization of the findings to the general population. Furthermore, this is the first study that evaluated the predictive value of both apoB/apoA-I ratio and CRP levels in relation to cancer mortality. The availability of detailed data on covariates allowed us to better control for confounding, for example, correcting our findings for dyslipidemia.

## 5. Conclusions

In conclusion, in the large representative samples of the US adult population, the apoB/apoA-I ratio and apoB levels significantly predicted cancer mortality, independently of several cardiometabolic risk factors. Furthermore, the predictive value of the apoB/apoA-I, but not apoB, to detect cancer death risk was better than CRP. If further evidence supports our findings, apoA-I and apoB measurements may be considered in general healthcare policies, especially for those at high risk of cancer development.

## Figures and Tables

**Table 1 jcm-09-00170-t001:** Characteristics of the study participants based on apolipoprotein B (apoB) levels to apolipoprotein A-I (apoA-I) ratio (3rd National Health and Nutrition Examination Survey (NHANES III)).

		Apolipoprotein B Levels to Apolipoprotein A-I Ratio				
		Q1 (*n* = 1923)	Q2 (*n* = 1925)	Q3 (*n* = 1924)	Q4 (*n* = 1923)	*p*-Value
Median (25th–75th)		0.50 (0.45–0.55)	0.65 (0.62–0.69)	0.80 (0.76–0.85)	1.01 (0.95–1.10)	
Age (Years)		44.1 ± 0.4	47.7 ± 0.4	51.2 ± 0.4	53.6 ± 0.4	<0.001
Sex	Men (%)	43.8	37.3	46.0	62.2	<0.001
	Women (%)	56.2	62.7	54.0	37.8	
Race/Ethnicity	Non-Hispanic White (%)	64.9	69.7	77.0	81.7	<0.001
	Non-Hispanic Black (%)	32.7	28.0	21.3	16.4	
	Mexican-American (%)	2.5	2.4	1.7	1.8	
Education level: <9th grade, (%)		19.9	24.8	28.0	28.6	<0.001
Poverty-to-income ratio (PIR) (n)		2.7 ± 0.03	2.2 ± 0.03	2.5 ± 0.02	2.6 ± 0.02	<0.001
Smoking (%)		25.1	26.9	28.4	26.1	<0.001
Alcohol consumption (g/day)		9.5 ± 0.2	7.4 ± 0.3	10.2 ± 0.2	15.1 ± 0.3	<0.001
Body mass index (kg/m^2^)		23.1 ± 0.3	25.1 ± 0.3	26.3 ± 0.3	27.1 ± 0.3	<0.001
Systolic blood pressure (mmHg)		118.1 ± 3.1	122.8 ± 2.9	124.1 ± 3.0	128.6 ± 2.8	<0.001
Diastolic blood pressure (mmHg)		70.4 ± 2.0	73.8 ± 2.6	74.4 ± 2.1	78.2 ± 2.1	<0.001
Fasting blood glucose (mg/dL)		118.2 ± 3.2	117.4 ± 2.8	119.3 ± 2.4	122.2 ± 2.5	<0.001
Triglyceride (mg/dL)		81.2 ± 2.3	94.4 ± 2.8	127.2 ± 2.4	178.5 ± 2.1	<0.001
Low-density lipoprotein (mg/dL)		109.2 ± 2.1	130.3 ± 2.0	143.4 ± 2.0	172.2 ± 2.4	<0.001
C-reactive protein (g/dL)		0.39 ± 0.02	0.42 ± 0.01	0.46 ± 0.01	0.45 ± 0.01	<0.001
Healthy Eating Index (HEI) (n)		63.2 ± 0.4	61.3 ± 0.3	63.2 ± 0.3	64.9 ± 0.4	<0.001
Dietary cholesterol (g/day)		216.1 ± 5.1	229.2 ± 4.5	235.4 ± 4.1	242.2 ± 5.2	<0.001
Energy intake (kcal/day)		1995.5 ± 12.1	2010.4 ± 13.9	2026 ± 11.5	2052 ± 12.4	<0.001
Carbohydrate (g/day)		219.2 ± 6.4	226.2 ± 4.8	220.3 ± 5.1	244.1 ± 6.2	<0.001

Groups across the quartiles were compared by either chi-squared test or analysis of variance. Values expressed as mean ± standard error of mean or %. NHANES: National Health and Nutrition Examination Surveys.

**Table 2 jcm-09-00170-t002:** Multivariable-adjusted hazard ratios (95% CI *) for cancer mortality across quartiles of apolipoprotein B levels to apolipoprotein A-I ratio (NHANES III).

		Apolipoprotein B Levels to Apolipoprotein A-I Ratio			
		Q2 (*n* = 1925)	Q3 (*n* = 1924)	Q4 (*n* = 1923)	*p*-Trend
Median (25th–75th)		0.65 (0.62–0.69)	0.80 (0.76–0.85)	1.01 (0.95–1.10)	
Cancer mortality	Model 1	1.08 (1.03–1.13)	1.44 (1.05–1.98)	1.62 (1.38–1.78)	0.002
	Model 2	1.10 (0.87–1.30)	1.35 (1.10–1.66)	1.40 (1.25–1.93)	0.005

Model 1: adjusted for age, sex, race, poverty-to-income ratio, and education level; Model 2: adjusted for age, sex, race, poverty-to-income ratio, education level, alcohol intake, Healthy Eating Index, smoking, body mass index, dyslipidemia, hypertension, and diabetes. * 95% CI: 95% confidence interval.

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
