# Peer review of "Apolipoprotein B/Apolipoprotein A-I Ratio Is a Better Predictor of Cancer Mortality Compared with C-Reactive Protein: Results from Two Multi-Ethnic US Populations"

_jcm, 2020, doi:10.3390/jcm9010170_

Round 1
Reviewer 1 Report
The work by Mazidi and coworkers aims to establish a correlation between ApoB levels and/or ApoB/ApoA-I ratio and cancer-related mortality in a large cohort of individuals with a follow-up of over 19 years. The study is well-conducted and the findings are interesting.
I have some suggestions and some issues that the Authors may want to address to strenghten the relevance and the quality of their work:
The Authors take into consideration two cohorts of individuals deriving from two subsequent studies: the NHANES-III (1988-1994) and the NHANES continuous (2005-2011). In the NHANES-III population they found 692 cancer-related deaths over a total of 7695 individuals with a mean follow-up of 19.1 years, corresponding to 8%. In the NHANES continuous population they found 402 cancer-related deaths with a mean follow-up of 4.1 years, corresponding to 5%. Although the follow-up time is very different in the two studies, it seems to me that the incidence of cancer-related mortality is higher in the NHANES continuous compared to NHANES-III. Is there a possible explanation?
The Authors demostrated that ApoB levels and ApoB/ApoA-I ratio are associated with increased risk of cancer-related mortality. Are these parameters also associated with mortality in general?
Due to the presence in literature of conflicting results on the association of ApoB/ApoA with different cancer types, is it possible that some cancer-types show a positive correlation and some others a negative correlation? Is it possible to stratify the patients by cancer type and see in which cancer type the correlation in maintained?
Please specify if NHANES-III or NHANES continous data are used in each Table/Figure
It seems that no display item (Table or Figure) is included relative to NHANES continous data described in lines 194-200 and relative to CRP correlation with cancer death (lines 202-208)
Line 278: please correct CD81 with CD8.
Reviewer 2 Report
In this study the authors investigated the value of apolipoprotein levels and C-reactive protein levels on cancer mortality. Using 2 large cohorts they find that the apoA/B ratio and the C-reactive protein level significantly predicted cancer mortality, albeit that the HR is rather low, even in the most optimal model.
Comments:
The title is confusing as it suggests that the ApoA/B ratio and C-reactive protein are comparable in predicting cancer, but the C-reactive protein was not associated with cancer mortality after adjustment for ApoA/B ratio. As such these parameters are interdependent and based on the multivariate analysis the ApoA/B ratio is clearly superior.
The ApoA and ApoB levels and changes thereof in relation to a particular cancer type seem to differ, as discussed by the authors. Since the author include all cancers, it is impossible to entangle this phenomena: if the trends are not the same for all cancers, the hazard ratios and confidence intervals become almost meaningless. For breast cancer patients for instance, this ratio may be meaningless, and the ratio may be much more meaningful for e.g. lung and colorectal patients. Discussion along these lines seem warranted.
The HR was attenuated when a number of known cancer risk factors (alcohol intake, smoking, BMI) were included (Model 1 versus model 2). Indeed, in the predictive comparisons the ApoA/B ratio remained significant even after adjustment for the risk factors that were included in model 2 (in fact the HR was very similar), suggesting that screening might be beneficial. Even though the data clearly show that individuals with higher ApoA/B ratios might have greater cancer mortality risk (with the caveat mentioned above), it is questionable whether this is translatable toward identification of individuals at risk with HR of 1.17.
Round 2
Reviewer 1 Report
The Authors answered all the raised issues
Reviewer 2 Report
The authors have revised the paper based on my comments. No other suggestions.